

# A geomorphic-process-based cellular automata model of colluvial wedge morphology and stratigraphy

Harrison J. Gray[1], Christopher B. DuRoss[2], Sylvia R. Nicovich[3], and Ryan D. Gold[2]

[1]US Geological Survey, Geosciences and Environmental Change Science Center, Lakewood CO 80225

[2]US Geological Survey, Natural Hazards Science Center, Golden, CO

[3]US Bureau of Reclamation, Technical Services Center, Lakewood CO 80225

*Correspondence to*: Harrison J. Gray (hgray@usgs.gov)

**Abstract.** The development of colluvial wedges at the base of fault scarps following normal-faulting earthquakes serves as a sedimentary record of paleoearthquakes and is thus crucial in assessing seismic hazard. Although there is a large body of

observations of colluvial wedge development, connecting this knowledge to the physics of sediment transport can open new frontiers in our understanding. To explore theoretical colluvial wedge evolution, we develop a cellular automata model driven by the production and disturbance (e.g. bioturbative reworking) of mobile regolith and fault scarp collapse. We consider both 90° and 60° dipping faults and allow the colluvial wedges to develop over 2,000 model years. By tracking sediment transport time, velocity, and provenance, we classify cells into analogs for the debris and wash sedimentary facies commonly described

in paleoseismic studies. High values of mobile regolith production and disturbance rates produce relatively larger and more wash facies dominated wedges, whereas lower values produced relatively smaller, debris facies dominated wedges. Higher lateral collapse rates lead to more debris facies relative to wash facies. Many of the modelled colluvial wedges fully developed within 2000 model years after the earthquake with many being much faster when process rates are high. Finally, for scenarios with the same amount of vertical displacement, different size colluvial wedges developed depending on the rates of geomorphic

processes and fault dip. A change in these variables, say by environmental change such as precipitation rates, could theoretically result in different colluvial wedge facies assemblages for the same characteristic earthquake rupture scenario. Finally, the stochastic nature of collapse events, when coupled with high disturbance, illustrate that multiple phases of colluvial deposition are theoretically possible for a single earthquake event.

## 1 Introduction

Characterizing the seismic hazard posed by major faults partly relies on understanding the history of prehistorical surface-rupturing earthquakes. To obtain this history, we must constrain the timing of past earthquakes. Various dating methods, such as luminescence and 14C dating, are commonly used to determine the age of stratigraphic and pedogenic units that pre- and



postdate an earthquake. The success of these dating methods often depends on how they are applied to sediments within a fault zone, particularly regarding stratigraphic location. One such postearthquake deposit, the fault-scarp-derived colluvial wedge (Malde, 1971; Swan et al., 1980; Schwartz and Coppersmith, 1984; McCalpin et al., 2009), is deposited immediately to thousands of years following fault rupture (e.g., Wallace, 1977). Colluvial wedges are typically found at the base of fault scarps (Figure 1) and are commonly used for reconstructing the history of earthquake events. As such, the paleoseismic analysis and interpretation of colluvial wedges directly feeds into seismic hazard assessments that in turn affect lives and livelihoods.

While there is a substantial body of literature on the classification and interpretation of colluvial wedges (see McCalpin, 2009; Chapter 3), there appears to be limited work connecting colluvial wedge development to process geomorphology, i.e. the mechanics of quantitative sediment transport, despite the importance of this deposit towards societal well-being. Specifically, we lack an ability to quantitatively predict the form and facies of colluvial wedges under varying environmental conditions, such as climate or lithology. We desire this predictive power so that we can develop knowledge toward understanding broader questions such as: 1) under what environmental conditions do you preserve (or not) a post-earthquake colluvial wedge; 2) are there conditions when a fault-scarp-generating earthquake does not produce a wedge, 3) How do these environmental conditions influence wedge morphology and internal stratigraphy; 4) are there geomorphic conditions that produce stratigraphy that can be misinterpreted as more than one earthquake event, and 5) what sort of time delay is expected between earthquake event and wedge formation? Here we propose a theoretical model of colluvial-wedge formation and stratigraphy that can provide the basis for quantitatively exploring these questions and improving our understanding of seismic hazards

## 1.1 Scope and Philosophy

The purpose of this investigation is to develop and evaluate a reduced complexity numerical model that can reproduce and explain (*sensu lato* Bokulich, 2013) major generalized colluvial wedge features such as wedge form over time and the typical distribution of sedimentary facies. Rather than attempting to simulate a specific colluvial wedge or field site, the numerical model proposed here is intended to explore model dynamics and conformity to field-based expectations and ultimately contribute toward the questions posed in the previous section. A secondary goal is to provide theoretical information on the development of colluvial wedges under varying conditions of mobile regolith (e.g. the inorganic fraction of soil) production, mobile regolith disturbance, and fault-scarp lateral collapse. Broadly, we are testing whether established geomorphic transport laws (e.g. Dietrich et al., 2003) generate synthetic colluvial wedges that agree with the contemporary understanding of colluvial wedge formation, specifically the Colluvial Wedge Conceptual Model (Nelson, 1992; McCalpin, 2009). If such laws cannot reproduce specific wedge features, additional geomorphic processes may need to be considered as explored in the discussion. Finally, on establishing agreement with the conceptual model, we offer testable predictions on the roles of our analysed geomorphic processes on colluvial wedge form and facies.



This work, as with many geomorphic models, is based on the modelling philosophy of using only the minimum level of physics (via established geomorphic transport laws; e.g. Dietrich et al., 2003) needed to reproduce a given class of geomorphic features

(e.g. Bokulich, 2013). As such, site-specific processes sometimes observed in field sites are not included in the model to maximize model generality. For example, we do not include more than one fault strand, back-tilting or rotation of the hanging wall (thus no fissure-fill deposits), more than a single faulting event, or sediment transport in and out of the model's 2-D plane. Furthermore, we have excluded non-colluvial geomorphic processes such as fluvial, lacustrine, or aeolian deposition. We also omit pedogenic processes such as the accumulation and translocation of organic matter, fine sediment, and pedogenic carbonate

because these vary widely across climate zones. The effects of these processes or how to quantitatively implement them in a model is not entirely clear and require further research. However, as explored in the discussion, the model presented here has significant utility for developing hypotheses and/or explanations of the sediment transport physics behind the colluvial wedge literature and in turn, our understanding of fault zone stratigraphy and earthquake hazard assessments.

## 1.2 Colluvial Wedge Morpho-Stratigraphy

Earthquake-related colluvial wedges typically consist of unconsolidated sediment deposited along the base of fault scarps with centimetre- to meter- scale vertical relief formed during surface-faulting earthquakes (Figure 1; McCalpin, 2003). Although colluvial wedges can form in any tectonic setting where a fault scarp is created (e.g., Nelson et al. 2014; Scharer et al., 2017), they are most commonly observed along normal-fault ruptures due to the creation of accommodation space and heightened preservation potential (e.g., Schwartz and Coppersmith, 1984; Machette et al., 1992; Galli et al., 2015; DuRoss et al., 2018;

Zellman et al., 2019). The term colluvial wedge is attributed to the wedge-like shape observed in profile and to the colluvial/gravitational processes that transport sediment (Schwartz and Coppersmith, 1984; McCalpin, 2009; Figure 2). Colluvial wedges reflect clastic sedimentation and pedogenic processes seconds to millennia following earthquake rupture and, depending on the parent material, typically consist of poorly sorted sediment and organic matter.

Observation of modern and ancient fault scarps allowed for the creation of a sedimentological facies and conceptual model for

the formation of colluvial wedges (Nelson, 1992; see review by McCalpin, 2009). The conceptual model envisions two general stages of deposition following a major earthquake that creates a fault scarp (Figures 1, 2). The first stage is the deposition of often poorly sorted material that occurs during or immediately after the earthquake when the exposed fault scarp destabilizes and collapses into a pile of irregularly bedded sediment and/or centimeter- to meter-scale coherent blocks at the base. The resulting deposit is given the classification *debris facies* by Nelson (1992) and McCalpin (2009), which is subdivided further

into categories such as the *upper debris facies* and *lower debris facies*. Following deposition of the initial debris facies, various sediment transport and mobile regolith-formation processes operate on the deposited sediment and exposed fault scarp, such as rain splash, bioturbation, and further gravitationally driven motion such as diffusive and granular flow (Wallace, 1977; Nash, 1980; Arrowsmith et al., 1994). These processes can modify the debris facies and/or induce further collapse on the



exposed fault scarp — also referred to as the 'free-face' by Wallace (1977) — and occur on the order of months to hundreds

of years after the earthquake. Generally, the lower debris facies include poorly sorted sediment, whereas the upper debris facies can demonstrate better sorting depending on sediment transport processes or the geometry of the lower debris facies pile. Finally, given enough time, the exposed fault scarp is sufficiently eroded/buried such that no further debris facies deposition can occur. Instead, the debris facies are buried by subsequent deposition, which results in the better sorted, finer-grained, and often stratified overlying *wash facies*. Wash and debris facies may develop soil profiles following sufficient passage of time

and associated surface stability.

## 1.3 Challenges in Modelling Colluvial Wedges

On fault scarps, the motions of sediment can be significantly larger than the averaging length scales needed to justify the use of continuity-based formulations (Furbish et al., 2018), such as diffusion-type equations often used to model fault scarp evolution (e.g. Colman and Watson, 1983). For example, a clast detached from the top of the scarp may fall and roll the entire

distance of the colluvial wedge, far in excess of local diffusive motion (e.g. Doane et al., 2019; Roth et al., 2020). Similarly, a disturbance brought on by fauna such as burrowing mammals may induce a granular collapse or mass failure leading to an ensemble movement of sediment not predicted by diffusion equations (Nash, 1984; Nash and Beaujon, 2006; Kogan and Bendick, 2011; Ferdowski et al., 2018). Finally, the processes occurring in this system act on timescales that span orders of magnitude, fractions of a second for a gravitational collapse, up to years or more for disturbance processes such as bioturbation

and mobile regolith production. Such a wide span of timescales poses a modelling challenge difficult to address with continuity-based formulations (Furbish and Doane, 2021).

A potential solution to the problem of non-diffusive conditions is the use of continuous-time cellular automata modelling (Murray and Paola, 1994; Tucker et al., 2016). This type of model is a computer simulation based around the idea of a grid consisting of individual cells. Each cell can have a unique state and can interact with other cells or transition into other cell

states based on user set rates and rules. The 'continuous-time' modifier indicates that all cell transitions are handled in a probabilistic manner and all cell transitions are computed to occur in an order set by the relative rates of transition processes. While cell state transitions occur between one or two cells at a time, a process with high transition probability may occur many times before a process with a low transition probability occurs. This allows us to create models that include processes with vastly different timescales such as gravitational fall versus mobile regolith production. Note that this treatment eliminates the

use of a 'time-step' and associated limitations common in continuum-style models.

Cellular automata models can be useful for simulating the movement of sediment, which often acts as a function of quantifiable sediment transport processes and the local topography (e.g. braided rivers: Murray and Paola, 1994; mobile regolith erosion: D'Ambrosio et al., 2001; also review by Ghosh et al., 2017). In the case of sediment transport, previous researchers have



created cellular automata models based around cell states such as air, stationary sediment, mobile sediment, and intact bedrock.
Cells can transition between each based on weathering and sediment transport rates leading to the accurate recreation of various landscape features not captured by continuum-style diffusion (Tucker et al., 2018; 2020). With cellular automata modelling, one can directly track the individual diverse motions of sediment with a level of detail not possible with continuum or diffusion style formulations. Likewise, whereas diffusion can approximate scarp erosion over millennia, it can't predict internal/subsurface wedge morphostratigraphy. Here, we apply cellular automata modelling to colluvial wedges created by
normal faults. Note that we did not explore the effects of multiple earthquake events or changes in recurrence interval on colluvial wedge formation because this topic deserves dedicated and separate study based on the foundations presented here.

## 2 Methods

### 2.1 Continuous Time Stochastic Cellular Automata Modelling

To model the formation of a colluvial wedge, we develop a continuous-time-stochastic cellular automata model using the
sediment transport physics from the GrainHill model (Figure 3) developed by Tucker et al. (2018), which is built from the CellLabCTS cellular automata framework (Tucker et al., 2016), itself a part of the Landlab modelling toolkit for the programming language Python 3.4 (Hobley et al., 2016; Barnhart et al., 2020). Within GrainHill, each cell type is programmed to interact with other cell types to recreate various sediment transport processes including gravitational collapse, momentum dissipation through elastic and frictional collision, mobile regolith production via weathering, and disturbance by mobile
regolith mixing processes. GrainHill can produce realistic sediment behavior such that one can model processes ranging from emptying of a grain silo up to generating characteristic forms of hillslope profiles (see Tucker et al., 2016; 2018; 2020 for a full analysis of the physics and sensitivity of GrainHill). The utility of this modelling framework allows for the modelling of specific geomorphic features such as colluvial wedges.

Within GrainHill, we generate a grid with hexagonal cells. Each cell within the model grid can consist of one of nine states.
State 0 is empty air, States 1-6 represent 'mobile regolith', with momentum in one of six directions, State 7 represents mobile regolith at rest, and State 8 is 'in-situ material,' or cells that represent uneroded parent material. Each mobile regolith cell is assumed to represent small aggregates of sedimentary material or individual clasts that can be mobilized by gravity, such as dry raveling, and/or mobile regolith disturbing processes, such as bioturbation. The in-situ material cells are abstractions used to represent parent material that is stationary unless a mobile regolith production process or a collapse process converts it into
mobile material. The in-situ material cells are not distinguished by lithology, such as between crystalline bedrock or consolidated sedimentary material (e.g., alluvium), instead assuming that differences between lithologies can be represented by differences in geomorphic process rates.





For our goal of a generalized model, we chose to focus on geomorphic processes that appear to be consistent across the majority of colluvial wedges, i.e. colluvial processes, while avoiding site-specific processes such as fluvial deposition. We find that the

minimum processes needed to produce colluvial wedges are mobile regolith production (rate given as $W_0$ with units of $yr^{-1}$), sometimes referred to as soil production in the geomorphology literature (e.g. Heimsath et al., 1997), mobile regolith disturbance (rate given as $D$ with units of $yr^{-1}$), roughly equivalent to 'soil diffusivity' (as frequently described in process geomorphology; Furbish et al., 2009; Tucker et al., 2016), and gravitational collapse (rate set by gravity $g$ converted to cellular automata transitions with units of $yr^{-1}$ per Tucker et al., 2018). The former two processes are well established geomorphic

transport laws in process geomorphology (Dietrich et al., 2003; Tucker et al., 2016). The latter process, gravitational collapse, is treated in a straightforward manner as described below. With this philosophy, we use the cell states, cell interactions (mobile regolith production and disturbance) and gravitational collapse from GrainHill. Furthermore, we extend the GrainHill gravitational collapse process to include lateral collapse, where an *in-situ* parent material cell can transition into a moving mobile regolith cell if it is laterally next to an air cell (rate of $LCR$ set as a fraction of gravity $g$ converted to cellular automata

transitions). Further discussion of non-included processes, such as pedogenesis, is in the discussion.

The model's initial conditions start (Figure 4) with a plane of in-situ material (Cell State 8) with a 5.71° initial slope (10% gradient) overlain by air cells (State 0). The cell size is set to 2.5 cm to attempt a balance between the resolution of the model run, computational time, grain size of coarse alluvial fan material common to normal fault zones and the approximate size of mobile regolith peds. This cell size allows for adequate resolution to model disturbance process acting throughout the thickness

of the mobile regolith. Alternate cell sizes (0.025–10 cm; electronic supplement) yield changes in the spatial resolution of the results but similar patterns of overall colluvial wedge morphology, transport, and velocity. We create a small sediment layer to simulate a pre-existing surface soil before inducing a single faulting event that produces a 2-meter tall fault scarp (Figure 4), consistent with historical earthquake surface ruptures (e.g., Crone et al., 1987; Caskey et al., 1996). Although normal faulting events can be larger, this scarp height allows us to capture detail in the resulting depositional features while keeping

computational demands reasonable.

Following the initialization, we model 2,000 years to capture the timescales of wedge formation (Wallace, 1977; McCalpin, 2009). Here, we model a single colluvial wedge to explore spatiotemporal trends in wedge formation and to avoid unnecessary complications related to repeated ruptures through older colluvial deposits. Note that the concept of a steady-state form is not applicable here as colluvial wedges are fundamentally transient features. Thus, a fixed model run time is needed and the

resulting deposits must be analysed with this in mind. We simulate both a vertical (90°) fault and a 60° dipping fault to capture two endmember colluvial wedge morphologies. The 60° endmember is consistent with Anderson's theory of faulting from tectonics (Anderson, 1951) and seismic observations (e.g., Jackson and White, 1989), whereas the 90° fault is based on the near-surface refraction of normal faults to steeper dips as commonly observed in paleoseismic exposures (e.g., Machette et al., 1992).



For the unconstrained parameters in mobile regolith production, mobile regolith disturbance, and gravitational collapse, we vary the rate of input parameters over four orders of magnitude and observe the resulting colluvial wedge form. We picked these magnitudes from the global range in mobile regolith diffusivity (i.e. Richardson et al., 2019) and converting this range into mobile regolith disturbance rate following equations in Tucker et al. (2018). To obtain a range for mobile regolith production rate, we note that the Peclet Number, a measure of the relative magnitude of mobile regolith disturbance versus

production, appears to fall into a range of 0.1-1 in global compilations, meaning that the orders of magnitude are roughly comparable (Gray et al., 2020). As such, we test the parameter space over a similar range as the mobile regolith disturbance rate. For the gravitational collapse, we let the mobile regolith cells overhanging air cells to collapse at a rate of free-fall following Tucker et al. (2016). For our lateral collapse process, we explored 14 orders of magnitude over which the rate had a visible effect on the modelled colluvial wedge form. From this large parameter space, we picked 4 orders of magnitude ($10^{-3}$g,

$10^{-5}$g, $10^{-9}$g, and $10^{-11}$g, where g is the rate of gravity in cellular transitions per time following Tucker, 2016).

## 2.2 Cell Tracking

Comparing the numerical model with real-world observations of colluvial wedge sediment is challenging as the mobile regolith cells do not record information such as sedimentary texture (i.e., grain size, grain shape, sorting, or clast orientation) that is typically used to distinguish between various sedimentary facies. An alternative is to classify the types and durations of

movements that occur to mobile regolith cells in the model and relate these as analogies to real world facies. For example, clasts within the debris facies of a colluvial wedge (Nelson et al, 1993) likely experience a greater transport in the vertical direction than in a horizontal direction, and transport likely occurs over a relatively short time. Conversely, sediment within the wash facies may be associated with greater overall horizontal transport than sediment in the debris facies, and may occur over prolonged time periods.

To explore spatial patterns of cell movement, we conceive of a transport index ($T_I$) to identify the movements and relative provenance of mobile regolith cells:

$$T_I = \frac{\Delta y}{\Delta x} \tag{1}$$

where $\Delta y$ is the total vertical distance a mobile regolith cell has travelled and $\Delta x$ represents the total horizontal distance, both following the scarp-forming earthquake. We chose this value as an index to evaluate mobile regolith cell motions because it

informs the viewer of the mobile regolith cell's overall path. Note that the cellular automata model used here has an inherent angle-of-repose of 30° due to the hexagon shape of the cells (Tucker et al., 2016). As such, cells with a transport index greater than $\sqrt{3}$ radians (due to special right triangle relationships) likely have spent more time in gravitational free fall than cells with values equal to or less than $\sqrt{3}$ radians. The latter being more likely to have traveled down an angle-of-repose slope.



Earth **Surface**
Dynamics
Discussions

EGU

Travel distance alone does not provide a complete picture of a mobile regolith cell's transport history. To further evaluate the
transport histories of mobile regolith cells, we measure the total time spent in transport and the average transport velocity and
give an example in Figure 7. The total transport time measures the total time a mobile regolith cell has been in a moving state,
which includes both gravitational free-fall and the episodic motions due to mobile regolith disturbance and the resulting mobile
regolith creep. The average transport velocity is the linear distance (calculated as $\sqrt{(\Delta x^2 + \Delta y^2)}$) divided by the total transport
time.

Next, we produce scatterplots of the various tracked metrics described above and plot them for each model run. In some cases,
the mobile regolith cells appear to form groupings based on their transport histories. We interpret these self-organized
groupings as analogs for various colluvial wedge sedimentary facies. Using the average transport velocity as a cutoff, we
classify cells with average transport velocity greater than $10^5$ m/yr as 'lower debris', cells with greater than 1 m/yr but less
than $10^5$ m/yr as 'upper debris', and cells with less than 1 m/yr as analogs for 'lower debris facies,' 'upper debris facies,' and
'wash facies.' Finally, we evaluate the effects of the geomorphic variables (mobile regolith production rate, mobile regolith
disturbance rate, and lateral collapse rate) on the colluvial wedge morphology and distribution of sedimentary facies.

**2.3 Sensitivity Analysis / Parameter Space Exploration**

We test the sensitivity of select parameters on the colluvial wedge morphology using the transport index, total transport time,
and average velocity as metrics. Since the number of parameters is large, the full parameter space requires excessive
computational time with many possible outcomes falling outside the realm of realistic geologic behavior. Instead we choose
to focus on the dip of the fault, mobile regolith disturbance rate, mobile regolith production rate, and lateral collapse rate, as
these appear to be key parameters controlling the colluvial wedge depositional environment. Parameters we keep constant are
the height of the scarp (2 m), the size of the hex cells (2.5 cm), the time of a model run (2 kyr) and the initial slope of the
faulted surface (5.7° / 10% slope). We also fix the GrainHill friction factor to an assumed value of 0.25 per Tucker (2018) to
include some elastic/momentum effects from inter-cobble collision noting that this does not appear to have a major impact on
the model results. An exploration of the fixed parameters would provide a useful perspective on the preservation potential of
various sized earthquake ruptures across different depositional environments. We leave these for future research as the goal
here is to obtain a general understanding of how colluvial sediment transport variables influence colluvial wedge stratigraphy.
The results of the parameter space exploration / sensitivity analysis are given in the supplemental material.

**3 Results**

**3.1 Modelled Colluvial Wedge Morphology**



Running the model across the parameter space produces a triangular deposit of mobile regolith cells located at the base of the modelled scarp (Figures 4,5; see supplemental material for full parameter space). During the initial stages of the run, the fault

face collapses and produces a small wedge of mobile regolith cells rapidly deposited within about a meter distance of the scarp. This rapid depositional phase is followed by a period of gradual deposition of mobile regolith cells, which eventually fill the available space between the top of the now-eroded scarp and the lower surface (Figure 4, Figure 5). The overall wedge for the 60° dipping fault is lower in total number of cells than the 90° fault scarp. The overall length of the modelled colluvial wedges are similar and both scarps both show similar levels of headward erosion in later timesteps of the models.

The overall displacement of individual cells is shown in Figure 6. Both 60° and 90° fault models show a greater amount of total horizontal movement for mobile regolith cells in the distal parts of the wedge versus the fault-proximal zone. The total vertical movement is similar with the distal parts of the wedge having mobile regolith cells with longer travel distances. The transport index, being a ratio of the vertical movement to the horizontal movement of a mobile regolith cell, shows a notable contrast between the 60° and vertical faults with the vertical fault model showing a much larger zone of high (>1.5) transport

index values. In contrast, the 60° fault model has a small zone of high transport index values mostly immediately adjacent to the fault. In both models, the zone of high transport index values is overlain by layers of mobile regolith cells with progressively lower transport index values that grade toward the surface of the wedge. After the colluvial wedge has filled the available accommodation space between the lower surface and the top of the fault scarp, there is an uppermost surficial layer of mobile regolith cells with consistent transport index values. The overall linear distance of transport (Figure 7A, 7B) is consistent with

the above observations.

A representative example of the temporal aspects of the path of individual cells is given in Figure 7. In both 60º and 90º fault models, there is a large wedge shape of mobile regolith cells with short (<1 yr) transport times overlain by a layer of cells with longer transport times (> 1-100 yr). The relative pattern of the transport times appears fairly consistent throughout the parameter space, although the details vary based on the geomorphic process rates. The patterns of the average transport

velocities generally consist of a wedge-shaped thin deposit of higher velocity cells overlain by a thicker layer of lower-velocity cells. Sometimes, a layer of low velocity cells is present within the overall high-velocity zone. The pattern of the average transport velocities across the parameter space can vary as the size of the higher velocity zone appears to increase with an increase in lateral collapse rate and mobile regolith production rate. Finally, we plot interpreted sedimentary facies on the model results using the classification criteria in the Methods section (Figure 8; Figure 9; Figure 10). Generally, the 60° dipping

fault creates a relatively small zone of debris facies overlain by wash facies of varying thickness. In contrast, the 90° fault is much more likely to create a relatively larger zone of debris facies.

## 3.2 Colluvial Wedge Sensitivity to Mobile Regolith Disturbance Rate, Mobile Regolith Production Rate, and Lateral Collapse Rate



Each of the three modelled geomorphic processes—mobile regolith disturbance, mobile regolith production, and lateral collapse—affect the morphology and stratigraphy of the resulting modelled colluvial wedge. First, the total area (number of mobile regolith cells) of the wedge represented by the model appears sensitive to all three parameters (Figure 11). Both mobile regolith production rate ($W_0$; x-axes in Figure 11) and lateral collapse rate ($LCR$; columns in Figure 11) share a positive relationship with colluvial wedge area for any collapse rate. Higher collapse rates broadly create larger wedges, but this effect appears limited by the amount of collapse-able *in-situ* parent material cells. The 90° fault appears to always create a larger colluvial wedge relative to the 60° fault, likely due to greater accommodation space.

The mobile regolith disturbance rate (D; y-axes in Figure 11), the process where mobile regolith cells can be randomly moved say by bioturbation (Figure 3), has a more nuanced affect: in model runs where the sediment supply is high due to raised mobile regolith production ($W_0$) or lateral collapse ($LCR$), an increase in mobile regolith disturbance will increase the total size of the colluvial wedge and expose *in-situ* parent material to further mobilization. This can be best observed in Figure 11 on the subplots where $W_0 = 10^{-3}$ yr$^{-1}$. In sediment-limited conditions such as lower mobile regolith production rate ($W_0$), higher mobile regolith disturbance rates can decrease colluvial wedge volume by mobilizing sediment away from the fault scarp. This effect is subtle but can be seen in the Figure 11 subplots when $W_0 < 10^{-5}$ yr$^{-1}$. Relatively high mobile regolith disturbance rates can eventually remove the entire wedge deposit if there is not mobile regolith production to replenish it and in some cases, the high mobile regolith disturbance rate works in tandem with high mobile regolith production to erode much of the fault scarp instead of lateral collapse (see extended results in supplemental material).

The ratio between the total area of the upper debris and lower debris facies (herein total debris facies) and the amount of the wash facies is sensitive to geomorphic variables (Figure 12). First, the 90° fault often creates a higher ratio of debris to wash facies than in the 60° fault, likely due to the greater number of collapse-able cells in the 90° fault case. Likewise, higher lateral collapse rates appear to increase the amount of debris relative to wash facies. One exception to this when lateral collapse rates are at moderate values for the 90° fault case (e.g. Figure 11G: $LCR = 10^{-5} g$ yr$^{-1}$) when the effects of mobile regolith disturbance and/or production create a relatively high amount of upper debris facies leading to a higher amounts of overall total debris facies.

Next, an increase in the mobile regolith disturbance rate and/or the production rate appear to increase the amount of wash facies relative to the total debris facies. This occurs due to the mobile regolith disturbance process reworking the post-collapse debris-like mobile regolith cells. In model runs with high mobile regolith disturbance relative to lateral collapse rates (e.g. Figure 11A,B,E,F), the total debris facies can be almost entirely reworked, thus producing cell transport histories meeting our 'wash facies' criteria. Finally, although subtle (Figure 11G,H), the mobile regolith production rate decreases the relative amount of debris by producing large relative amounts of mobile regolith that then travel down the scarp into the colluvial wedge.



## 4 Discussion

Our primary goal in this study is to evaluate model results to infer the theoretical effects of geomorphic process rates on colluvial wedge stratigraphy. However, the model here does have limited — but useful — explanatory power (*sensu* Bokulich, 2011) in that it provides a connection between the field-based knowledge on conceptual colluvial wedge formation with the physics-based principles of sediment transport from process-geomorphology. First, we must evaluate the realism of the model with respect to observations of colluvial wedge formation, both modern and observed in paleoseismic trenching of fault zones. Next, we must confirm if our choice of classification of mobile regolith cells into sedimentological facies designations is accurate, and if they produce patterns that match our conceptual model of colluvial wedge formation. After this, we will describe the implications for colluvial wedge formation and interpretations. Note that here, *sediment* refers to real-world colluvial wedge material and mobile regolith refers to the modelled cells. While technically we are discussing the transport histories of mobile regolith *cell states*, we will refer to them as mobile regolith *cells* for colloquiality even though the cells themselves are stationary.

### 4.1 Model Realism

The most basic validation one can make is that the model resembles colluvial wedges observed along historic and prehistoric fault scarps, those revealed in paleoseismic exposures, and the idealized form in the colluvial wedge conceptual model. The profile of the developed scarp is similar in appearance to modern examples of colluvial wedge formation, such as observed following the 1983 Borah Peak rupture (Crone et al., 1987; McCalpin et al., 1995; e.g., Figure 2 versus Figure 4, Figures S1–S8). The full parameter space we explore produces fault scarps that range from essentially no wedge development up to almost complete erosion of the fault scarp, although we focus on runs most resembling real colluvial wedges (Figures 9 and 10). The modelled wedges here bridge much of the gap between the classic wedge-shaped forms of the colluvial wedges (Wallace, 1977; Nelson, 1992; McCalpin, 2009, their Figure 4.11) and forms that resemble a continuous mobile regolith layer with active transport (e.g., Bennett et al., 2018; DuRoss et al., 2018; Gray et al, 2019). We argue that the agreement between the model and previous theory and observations is evidence that the model is providing a reasonable analog of colluvial wedge development. Although the resemblance to real/idealized colluvial wedges does not *prove* that the underlying model mechanics are correct; the use of established geomorphic transport laws to reproduce the idealized forms described in the colluvial wedge conceptual model (Nelson, 1992; McCalpin, 2009) provides a mechanistic explanation for colluvial wedge formation and morphology.

Next, we must assess if the transport of mobile regolith cells conforms to field observations and previous theory. First, as noted in the results, the transport histories of mobile regolith cells form groupings that are not explicitly coded into the model (Figure 8). These groupings arise because of the sediment transport physics from established geomorphic transport laws and/or the interactions of the cells as the model run progresses. Because these groupings appear distinct, we classify them into analogs for the apparently similar sedimentary facies defined in colluvial wedge literature (e.g. Nelson, 1992). An analysis of our



sedimentary facies interpretations is provided in Section 4.3 below. The grouping of mobile regolith cells we classify into lower debris facies (Figure 8) appear derived from relatively proximal sources (higher $T_I$ values, lower linear distance values), such as material from the fault scarp, and have overall short transport times. We suggest this would match the poorly sorted

sediment of real-world lower debris facies. The grouping of cells we classify into the upper debris facies show similarly proximal provenance, but longer overall transport times (Figure 8). This upper debris facies involves both material from collapse and headward erosion of the scarp and reworking of the previously deposited lower debris facies. It should be noted that both upper debris and lower debris facies do not always occur subsequently and that interbedded layers of either, sometimes even including wash-like layers, can occur as a function of the relative geomorphic process rates. (Figures 9, 10).

This observation possibly gives a mechanistic explanation for the complex stratigraphy observed in actual colluvial wedges. Finally, the grouping we classify into wash facies is represented by distal provenance, long transport times, and low average velocity of transport (Figure 8). Presumably, the effects of the transport histories represented by the wash facies analog cells would result in the better-sorted and finer grained sediment observed in real colluvial wedges wash facies. The match-up between our modelled facies and real-world facies suggests agreement between the model and interpretations of colluvial

wedge development through sedimentary analysis (Nelson, 1992; McCalpin et al., 1993; McCalpin, 2009).

The timescales of modelled versus actual colluvial wedge development must also be considered when assessing model realism. First, many of the model runs with higher collapse rates produce lower and upper debris facies cells rapidly with some developing in approximately ≤10 years (Figure 10G, 10H). At the highest collapse rates, the fault scarp fails immediately after the formation of the fault scarp consistent with observations of modern wedges (Figure 1; Wallace, 1977; McCalpin, 1993).

This initial collapse phase is followed by a longer period of deposition (up until the end of the model run of 2000 years) and is associated with longer-term processes such as reworking of debris facies and production of the rest of the wash facies (Figure 7C, D). These timescales appear to broadly match expectations of wedge development (Forman et al., 1988; 1991; McCalpin, 2009). When collapse rates are lower, or at least of comparable magnitudes to mobile regolith disturbance rates, more complex stratigraphy can be produced with layers of cells matching upper or lower debris facies criteria. Some model runs appear to

show layering from individual collapse events interspersed with reworked mobile regolith cells (Figure 10B, 10D). Such a system appears to create stratigraphy that may resemble multiple small earthquakes rather than one large event. This result may imply that the stochastic nature of collapse means that theoretically, a single earthquake event could produce apparently multi-earthquake stratigraphy if geomorphic process rates allow, at least within the domain of our model conditions.

From these observations, we argue that this model can reasonably reproduce colluvial wedge morphologies and gross
sedimentary facies over realistic time frames. Furthermore, the model we describe here offers a way to connect colluvial wedge theory with the first-principles physics of sediment transport. We conclude that the model can be used to hypothesize how changes in geomorphic variables affect colluvial wedge development with some limited mechanistic explanatory power for the idealized colluvial wedge conceptual model.



## 4.2 Sensitivity to Geomorphic Parameters

Our results suggest that theoretically, geomorphic variables (i.e. mobile regolith production rate, mobile regolith disturbance rate, and lateral (free face) collapse rate) directly impact colluvial wedge form and sedimentology. The most significant theoretical result from this study is that facies distributions may not necessarily occur in a sequential order. Interbedding of layers with different transport histories is possible, particularly when collapse rates are low (Figures 9, 10). Other researchers (e.g. Gray et al., 2019) have noted site-specific field relations that suggest greater complexity behaviour in real colluvial

wedges than is currently represented in the colluvial wedge conceptual model. A possible consequence of complex behaviour is that a single faulting event could theoretically produce multiple facies sequences that resemble discrete faulting events, despite only a single event occurring. Whether this happens in real-world situations remains to be tested, but the possibility may complicate paleoseismic hazard assessments. However, this may be avoided by soil profile and geochronological interpretations (e.g. Berry, 1990).

Other theoretical effects of geomorphic parameters match interpretations present in the literature. For example, the debris facies in the model can form either due to the lack of internal cohesion of the parent material (high rates of lateral collapse), or due to mobile regolith forming processes acting on the exposed free-face, similar to interpretations by Wallace (1977). Such mobile regolith forming/disturbing processes could involve burrowing mammals, root growth, shrink-swell, and freeze-thaw cycles among many others (Gray et al., 2020). Another example is the field-observed effect of microclimate on scarp

degradation where aspect-controlled water contents affect the degradation rate (e.g. Pierce and Colman, 1986; Pelletier et al., 2006). In the model, such variance in degradation rate can be reproduced by varying the relative geomorphic parameters noting that the specific recreation of a field-site is beyond the scope of the model.

Additionally, depending on the rates of geomorphic processes, the colluvial wedge could theoretically undergo substantial reworking that in turn affects information on earthquake timing such as that recorded by geochronology. For example, model

runs with high relative mobile regolith disturbance rates appear to substantially rework the initially deposited debris facies, converting some fraction of it into wash facies over time. When mobile regolith disturbance rates sufficiently exceed mobile regolith production and lateral collapse rates, the entire wedge can be converted into the wash facies classification given enough time. One could reasonably surmise that geochronometers such as [14]C and OSL would produce younger ages via incorporation of more recent carbon and sunlight exposure. The extent to which this happens in nature is debatable. Generally,

mobile regolith disturbance processes are linked with mobile regolith production processes (Wilkinson et al., 2009; Roering et al., 2010) such that the relative magnitude of both disturbance and production appear relatively constant across climate zones (Gray et al., 2020). However, one could imagine a fault scarp in highly cohesive material (low collapse rates) in an arid climate with low mobile regolith production rates, but with high disturbance rates with burrowing desert fauna. Such a case may be testable with a meta-analysis of geochronology from paleoseismic studies.





As the rates of mobile regolith production and mobile regolith disturbance play a large role in colluvial wedge form, we should
acknowledge when such values are expected to be high versus low. There is evidence that wetter climates with higher mean
annual precipitation can have higher mobile regolith disturbance rates than drier climates assuming that mobile regolith
disturbance is coincident with mobile regolith transport rates (see data in Richardson et al., 2019: their Figure 2). Similarly,
higher mobile regolith disturbance rates may be associated with locations of higher mobile regolith production rates because
the processes that disturb mobile regolith are often the same or closely related to the processes that create mobile regolith
(Gray et al., 2020). Based on the model results, the wash-dominated colluvial wedges created with higher mobile regolith
disturbance/production rates may be associated with wetter climates including microclimates (Pierce and Colman, 1986;
Pelletier et al., 2006). Conversely, our model runs that produce debris-dominated colluvial wedges are often associated with
lower values of mobile regolith production/disturbance. A reasonable hypothesis following these observations is that climate
may impact the sedimentology of a colluvial wedge for a given lithology. Consequently, an earthquake may produce a different
colluvial wedge in a wetter climate versus a drier climate for the same given amount of displacement with all other variables
held constant.

Finally, it is not just geomorphic variables that control colluvial wedge form. It is important to note that the tectonically-
controlled angle of the fault scarp appears to increase the sensitivity of the relationships between the geomorphic variables and
the colluvial wedge morpho-stratigraphy (Figures 10, 11). For example, the 90° fault contains a large propensity for collapse,
whereas the gentler slopes of the 60° fault do not. Mobile regolith cells in the 60° fault tend to episodically tumble downslope
via gravity, the overall rate at which depends on the presence of other mobile regolith cells on the fault scarp. If there are a
significant number of mobile regolith cells on the slope, this can slow down the overall transport of a cell state. This impedes
the rapid deposition of debris facies and causes the 60° fault to more likely consist of wash and lower debris facies cells
(Figures 9, 10), whereas the 90° fault hosts both wash, upper debris, and lower debris facies. As a final note, the steeper 90°
fault generally creates taller and thicker colluvial wedges, likely due to the greater accommodation space created by the tall
scarp, which suggests that estimates of fault displacement based on wedge thickness alone (e.g., Ostenaa, 1984; Klinger et al.,
2003; McCalpin, 2009; Bennett et al., 2018; DuRoss et al., 2014, 2018) could be complicated by near-surface fault geometry.

**4.3 Comparisons between the Model Facies and Actual Sedimentary Facies**

To make comparisons between the model and real colluvial wedge stratigraphy, we needed to first classify which mobile
regolith cells are analogous to the various sedimentary facies in colluvial wedges. In this study, we chose classification criteria
based on average transport velocity to divide mobile regolith cells into those resembling upper debris, lower debris, and wash
facies. Although other subdivisions are possible with our model, such as those in the colluvial wedge conceptual model (e.g.,
Nelson, 1992), we did not pursue those to limit the influence of subjective criteria in our interpretations and to stay close to
the general concepts of colluvial wedge development. The classification criteria we chose are based on the self-organized
groupings of cells transport histories from the model runs (Figure 8 and supplemental material). Admittedly, the sharp cut-offs





in velocity value for each facies do not capture the apparently diffuse boundary between groupings seen in the scatter plots (Figure 8). A potentially better method may be to use a statistical tool such as a mixing model, but it is beyond the scope of this study to explore this point further. However, we argue that the facies designations are reasonable boundaries for the self-

organized groupings and allow us to use the model as a general analog for colluvial wedge morpho-stratigraphy

The modelled upper and lower debris facies largely resemble those described by the standard colluvial wedge facies model (Nelson, 1992). The modelled debris facies create a wedge-shaped zone that increases in thickness with proximity to the fault (Figures 8,9, S33-S40) similarly to natural debris facies. This initial deposit of colluvium helps preserve the original dip of the fault plane (exposed fault free-face), which is steepest adjacent to the debris facies but becomes progressively more eroded

and gently dipping where buried by wash facies sediment. As the cells in the model are uniform and there is not a clear way to model grain sorting, our separation between upper/lower debris facies in the model is largely based on transport characteristics. There are however similarities between the formation of the upper/lower debris facies in the model and hypothesized forming mechanisms reported by field-based studies (Wallace, 1977, Pierce and Colman, 1987, Nelson, 1992, McCalpin, 2009), namely that both lateral collapse of the free-face and collapse due to mobile regolith production/disturbance

(Figure 12) can occur by the same processes. An example may be bioturbation both mixing soil and also inducing collapse via burrowing into the fault free-face. Finally, there is agreement with the maximum timescales of the model's formation of the debris facies (<100 to <1000 years) and observations of modern to historical fault scarps (Wallace, 1977, McCalpin, 2009).

The modelled wash facies also have features that simulate their natural analog. First, the shape of the wash facies spans from very thin layers overlying the debris facies, up to large elongate layers that reach their maximum thickness near the toe of the

debris facies (Figures 8, 9, S33-S40). In contrast to the modelled debris facies, the modelled wash facies tend to be constantly reworked and have lower average transport velocities and longer transport times. In many model runs, a persistent layer of active sediment transport exists near the surface of the wash facies similar to observed field relations (McCalpin, 2009; Gray et al., 2019). This persistent reworking by mobile regolith disturbance is analogous to the current model of wash facies development by bioturbation and sheet-wash processes (Nelson, 1992).

**4.4 Implications and Conclusions for Colluvial Wedge Formation and Interpretations**

The methods and results presented here offer three implications to the state of knowledge on colluvial wedge development. The first is that the model can connect the observations of a wide variety of natural colluvial wedge morphology into a physics-based model that appears to accurately reproduce a large variety of colluvial wedge development. This means that despite the wide range in geomorphic mobile regolith production and disturbance processes at play in colluvial wedge environments, our

general theory for their development (Wallace, 1977, Nelson, 1992, McCalpin, 2009) seems to be largely accurate. Although this is not a formal test of the colluvial wedge conceptual model (Wallace, 1977; Schwartz and Coppersmith, 1984; McCalpin, 2009), it provides support to the idea that observations of modern wedge development can be extrapolated into a theory of





colluvial wedge development over geomorphic timescales using geomorphic transport laws (Nelson, 1992; Dietrich et al.,
2003). A corollary of this implication is that the effects of just a handful of abstracted geomorphic processes (here, mobile
regolith production rate, mobile regolith disturbance rate, and lateral collapse rate) can explain a wide variety of colluvial
wedge forms. Although these parameters yield realistic wedges, testing how wedge morpho-stratigraphy responds to more
complicated factors such as steep and/or variable surface topography, complex (e.g., distributed and antithetic) faulting, and
repeated surface ruptures through time would further build confidence in relating physics-based models to natural colluvial
wedge observations.

The second implication of this study is that the effects of fault plane angle, mobile regolith production, mobile regolith
disturbance, and free face collapse are important to consider when interpreting a sequence of colluvial wedge formation. The
angle of the fault plane appears to play a large role, with the steeper dipping fault planes leading towards faster wedge
development (Figure 7), a larger proportion of debris facies (Figures 8, 9, 12), and larger (thicker and more laterally extensive)
wedge deposits (Figures 8, 9, 10). As a result, in the case of repeated fault rupture, colluvial wedges along steeply dipping
faults may be more likely to be identified and correctly interpreted as evidence of fault rupture than those along lower angle
faults, which could be more difficult to differentiate given their reduced volumes. The higher rates of lateral collapse, say for
unconsolidated parent material, correlate with the rapid development of debris facies (Figure, 12). However, high rates of
mobile regolith production/disturbance on parent material with low collapse rates can also generate debris facies, although
likely in a slower fashion.

Finally, the combined effects of the rates of mobile regolith production and disturbance can theoretically create interbedding
in colluvial wedges. Multiple episodes of collapse and reworking can theoretically occur for a single earthquake event, although
it seems like this would become less likely through time as the free face is progressively eroded and thus the source of collapse
failure diminished. Additionally, earthquake events with similar magnitude and displacement can have different colluvial
wedges if the mobile regolith production/disturbance rates are different, say from a difference in climate or environment. This
idea could be explored with models paired with meta-analysis of colluvial wedge shapes, sizes, and strata formed in different
climates from the published literature (N. Reitman, in-person communication). We hypothesize that geomorphic process rates
control the extent of reworking, which may in turn affect the results of geochronology. Finally, as mentioned in the
introduction, the model presented here offers a potential new means to explore further questions on colluvial wedge
development over timescales longer than the modern record.

**Acknowledgments, Samples, and Data**

The model code used to generate the results on this paper will be uploaded to the Community Surface Dynamics Model System
repository. This work was partially supported by the U.S. Geological Survey Earthquake Hazards Program. Any use of trade,
product, or firm names is for descriptive purposes only and does not imply endorsement by the U.S. Government.



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





**Figure 1.** Illustration of the surface expression of a colluvial wedge forming environment. (A-B) ~2-3-m high surface rupture associated with the 1983 M6.9 Borah Peak, Idaho earthquake. Photographs show initial debris facies (DF) colluvium deposited along the base of the fault scarp and exposed fault free face (FF). Rod in (A) is 2.8 m high; photographs taken near Doublespring Pass road in 1983 and are available at https://library.usgs.gov/photo/#/?terms=Borah%20Peak (C-D) Similar fault scarps near Doublespring Pass road photographed in 2015. Person in (D) is ~ 1.5 m tall. Photographs C-D taken May 2015 by C. DuRoss.





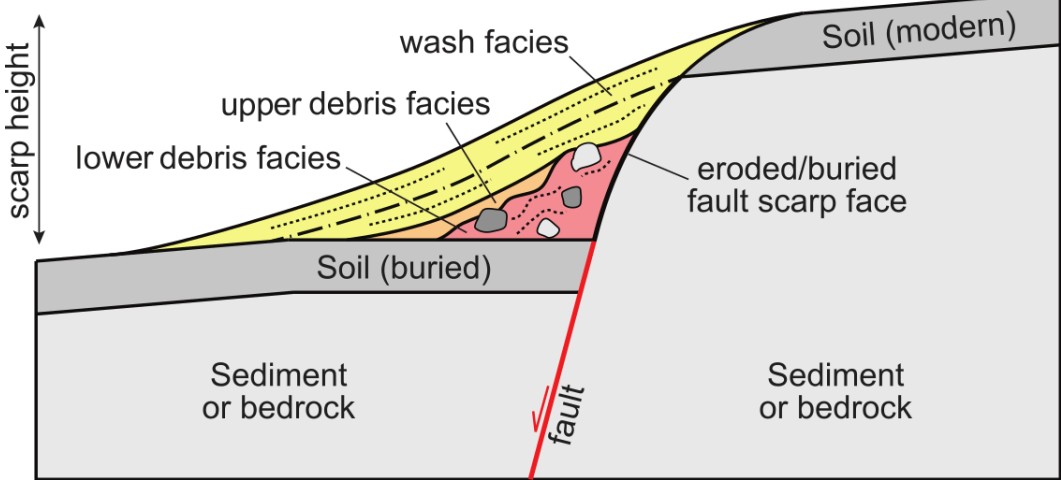

**Figure 2.** Schematic of the colluvial wedge conceptual model describing morphology and stratigraphy. The lower debris facies represent sediment deposited by initial collapse of the fault scarp, whereas the upper debris and the wash facies and overlying colluvium represents deposition over longer timescales. Terminology adapted from Nelson (1992) and McCalpin (2009).





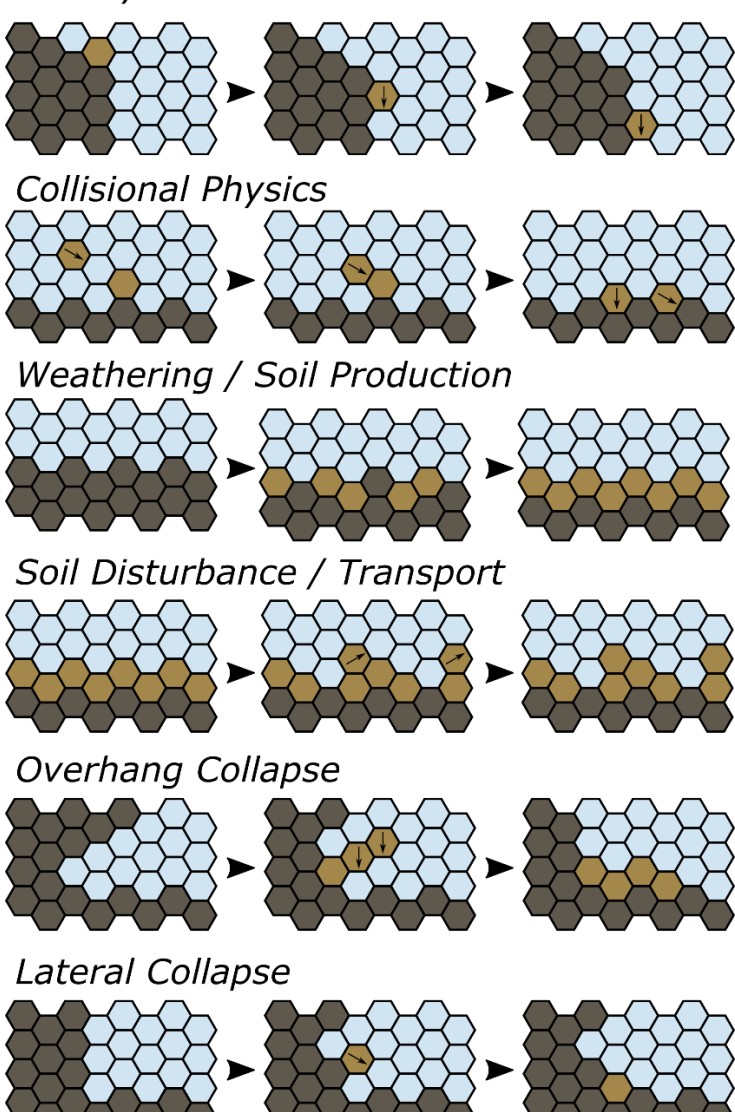


**Figure 3**. Illustration of a "cellular automata" type model used to simulate colluvial wedge formation. Figure shows geomorphic processes included in the model for air, mobile regolith, and in-situ parent material cells (following Tucker et al., 2016;2018;2020). Definitions for "mobile regolith" and "in-situ parent material" are given in Section 2.1.





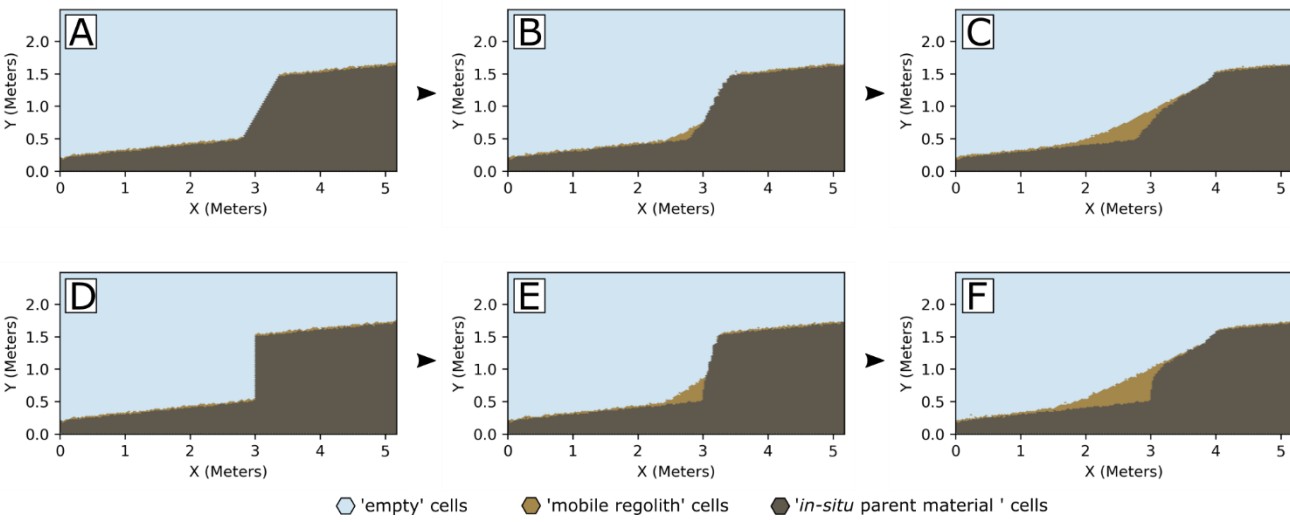


**Figure 4:** Example of the evolution of the colluvial wedge cellular automata models for a 60° dipping fault (A, B, C) and for a vertical fault (D, E, F). In both models, there is an initial collapse where the fault face produces a layer of rapidly deposited mobile regolith cells that create a small wedge with a slope approximate to the angle of repose (30°). The collapse phase is followed by a period of gradual deposition of mobile regolith cells that create an overall elongate wedge.






Earth **Surface**
**Dynamics**
Discussions

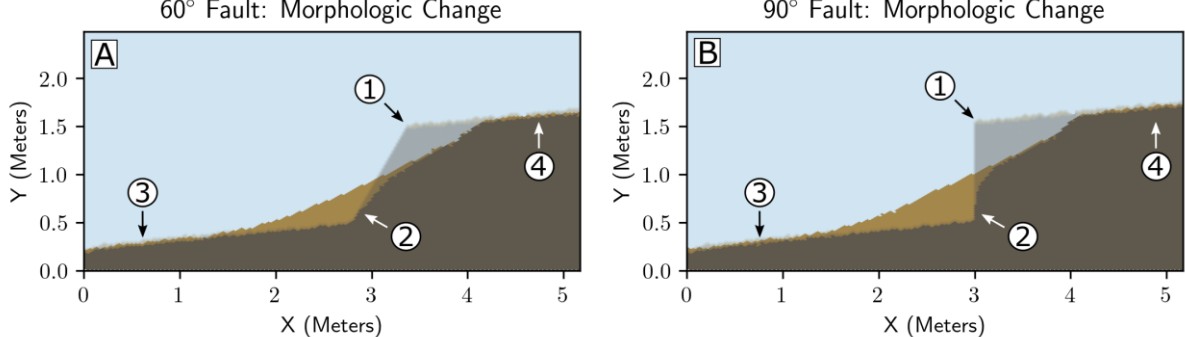

**Figure 5:** Example of the change in fault scarp morphology and colluvial wedge development in both the (A) 60°
dipping fault and (B) 90° dipping fault. The transparent zone shows the initial fault scarp. See Figures 3,4 for
legend. 1) The upper free-face of the scarp erodes headward by both lateral collapse and by erosion from mobile
regolith production and mobile regolith disturbance processes. A small knickpoint-like ledge may be visible if the
scarp has not fully diffused. 2) A small trace of the fault free-face is buried by the colluvial wedge whereas the
fault trace higher above has been eroded.  The colluvial wedge volume is larger for the 90° dipping fault due to the
increased accommodation space. 3) The mobile regolith on the hanging wall is partly buried by the colluvial wedge
whereas the still-exposed areas continue to develop over time. 4) The mobile regolith forming processes on the
upthrown footwall continue and produce sediment that migrates downhill and deposit onto the colluvial wedge.



Earth **Surface**
**Dynamics**
Discussions

**Figure 6:** Representative illustration of the horizontal and vertical displacements of the mobile regolith cells from the sensitivity analysis. A,B) Total horizontal displacement defined as the change in x-position of a cell from the earthquake event until the end of the model run, for the 60° fault (A) and the vertical fault (B). C,D) Equivalent plots for the total change in the vertical y-position. E,F) Plots of the ratio (the transport index) between the total vertical movement and total horizontal movement.



Earth **Surface**
**Dynamics**
Discussions

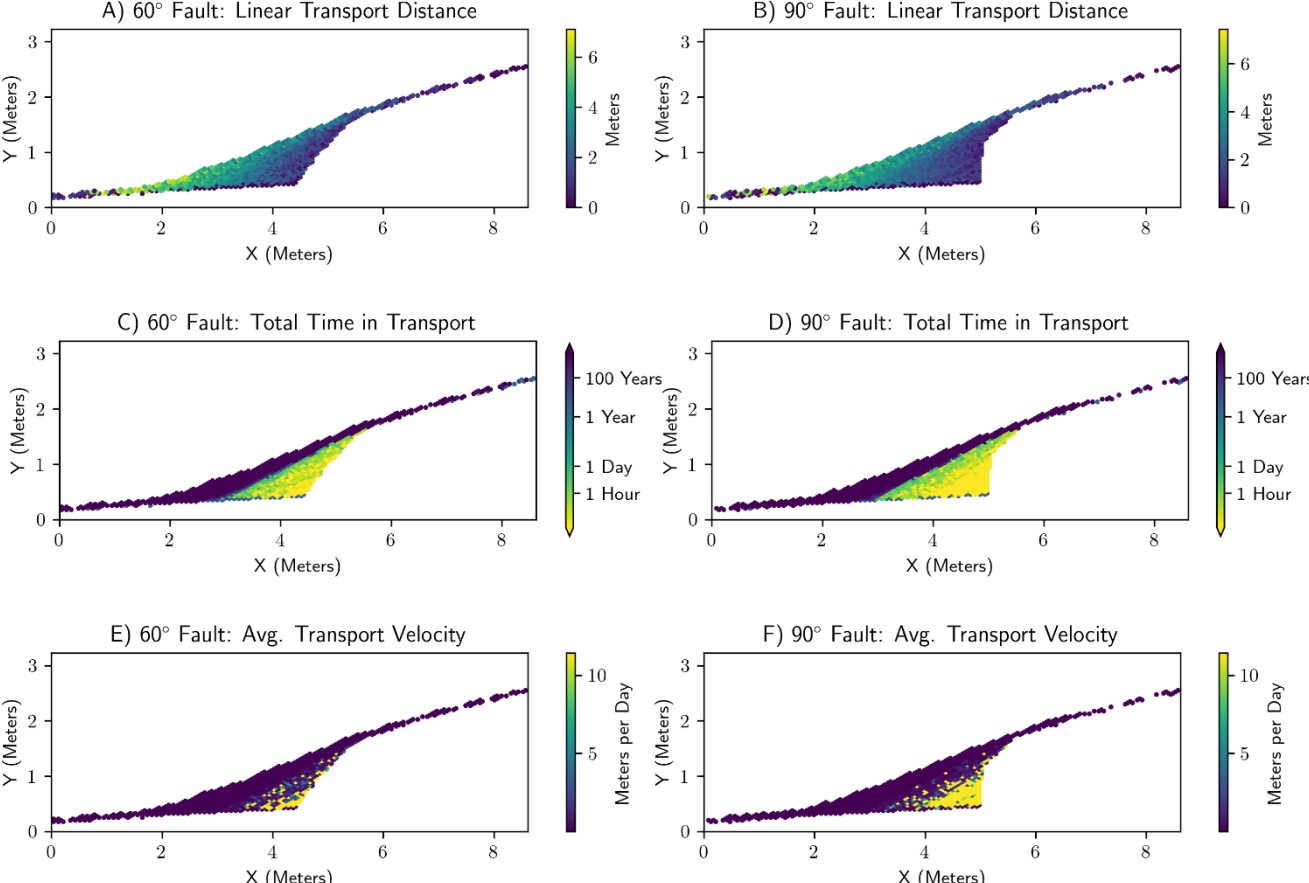


**Figure 7:** Illustration of the rates of mobile regolith cell transport. A,B) The linear distance from the mobile regolith cell's original position at the start of the model run (linear distance $= \sqrt{\Delta x^2 + \Delta y^2}$) for the 60° (A) and 90° faults (B). Generally, the fault-distal parts of the wedge involved further-travelled mobile regolith cells. C, D) The total time a mobile regolith cell spends in transport before coming to rest and burial. For both the 60° (A) and 90° faults

(B), the colluvial wedge appears to have mostly developed within 1-10 model years. Some model runs with high relative collapse rates can produce a rapid initial deposit within hours to days. This rapid initial deposit is buried relatively slowly over 100's of years until the modelled colluvial wedge reaches its maximum volume. E, F) The average velocity of transport, calculated as the linear distance (A,B) divided by the total transport time (C,D). The higher average velocity indicates that the cell has travelled a significant distance in a relatively short period of time

(e.g., gravitational free fall) compared to those that travel similar or shorter distances over a longer period of time (e.g., mobile regolith creep). Note higher number of high-velocity cells in the 90° versus the 60° fault.



Earth **Surface**
**Dynamics**
Discussions

**Figure 8:** Scatter plots of the various transport histories of mobile regolith cells (Total Transport Time, Transport Index ($\Delta y/\Delta x$), and average transport velocity). A, B, C, D) Transport time versus transport Index for various lateral collapse rates. E, F, G, H) Transport Time versus Linear Distance for various lateral collapse rates. I, J, K, L) Average transport velocity versus Transport index. The scatter plots appear to show natural groupings of cells with similar transport histories. The color indicates our interpreted groupings of cells into analogs for the various sedimentary facies in the colluvial wedge model. These groupings are arbitrarily divided by the transport velocity with values greater than $10^5$ m / yr being 'lower debris' analog facies and cells with values lower than 1 m / yr being 'wash' facies analogs. Cells between these values are classified as 'upper debris' facies analogs.



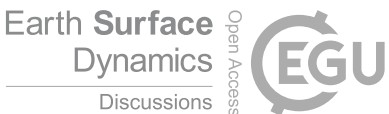

Earth **Surface**
**Dynamics**
Discussions

**Figure 9:** Example illustration of modelled colluvial wedges and facies analogs for 60° and 90° faults per varied geomorphic process rates with a fixed lateral collapse rate of $10^{-5}g$ (see Figure 3: D = disturbance rate, $W_0$ = mobile regolith production rate). Order of magnitude changes in the geomorphic process rates result in different wedge forms and stratigraphy.





**Figure 10:** Example illustration of modelled colluvial wedge with sediment facies analogs for 60 degree and vertical fault planes as a function of lateral collapse rate (LCR). The lateral collapse rate is taken as a fraction of gravity, e.g. "LCR = 1e-05" is a rate of $10^{-5}$ times gravity entered into the lateral collapse function (Figure 3). Higher collapse rates generally promote larger colluvial wedges and larger volumes of debris facies versus wash facies for a given geomorphic process rates (D, $W_0$).



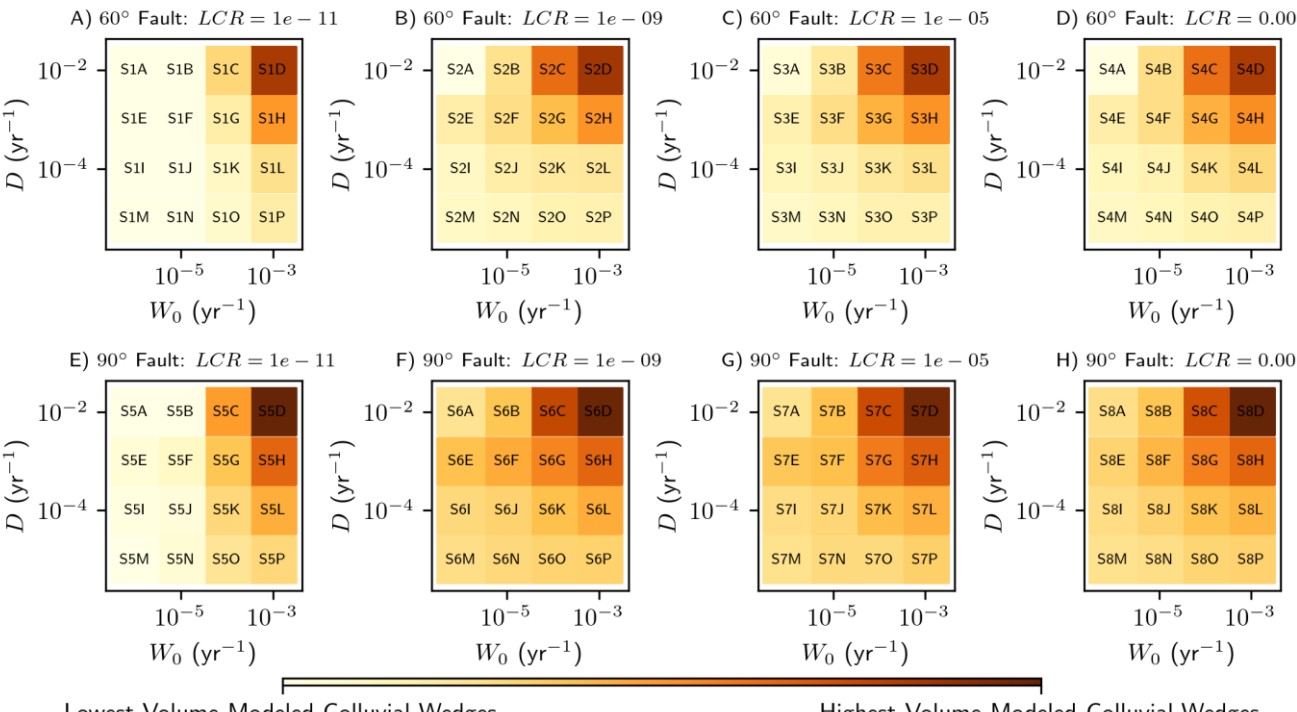

**Figure 11:** Illustration of the influence of mobile regolith Production rate ($W_0$, horizontal axis of each plot), mobile regolith Disturbance Rate ($D$, vertical axis of each plot), and Lateral Collapse Rate ($LCR$, increasing from left to right), and fault dip (60° vs. 90°) on the total volume of the resulting colluvial wedge from a 1 meter tall fault scarp after 2 kyr simulation time. Broadly, the Mobile regolith Production rate and Lateral Collapse Rate control the overall sediment supply and the Mobile regolith Disturbance Rate controls the rate of transport of sediment. Some nuance exists such that higher disturbance rates or lateral collapse rates expose bedrock and facilitate mobile regolith production. Finally, fault dip appears to control the overall accommodation space available, with steeper faults producing larger colluvial wedges. Number codes within plots indicate figure number in the supplemental material. For example, code 'S6F' refers to Supplemental Figure S6, subplot F.

Earth **Surface**
**Dynamics**
Discussions

715

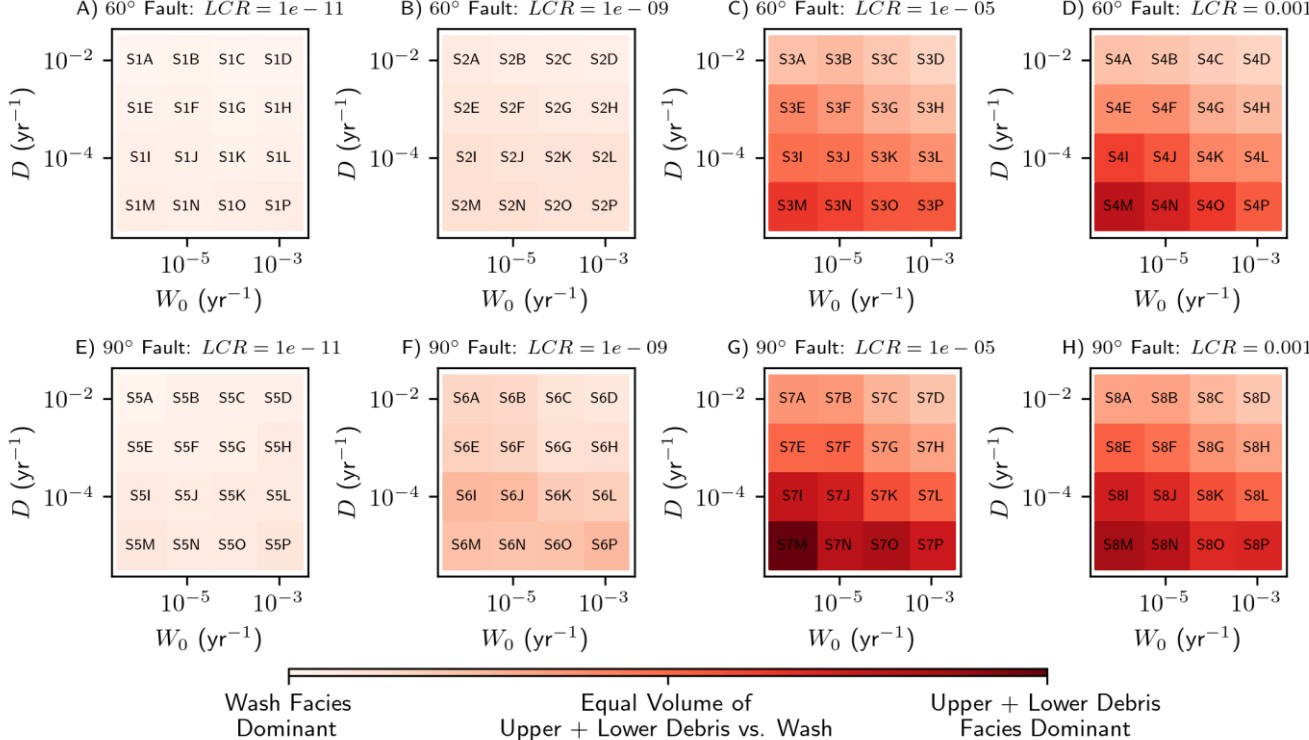

**Figure 12:** Illustration of the effects of geomorphic variables (see Figure 10 caption) on the ratio of Debris facies volume versus the volume of the wash facies. A value of 1.0 represents equal proportions of debris facies volume to wash facies volume. Number codes within plots indicate figure number in the supplemental material. For

720    example, code 'S38F' refers to Supplemental Figure S38, subplot F.