# Peer review of "A geomorphic-process-based cellular automata model of colluvial wedge morphology and stratigraphy"

_Earth Surface Dynamics, 2021_

## Author Comment (AC1)

**IRC1**: 'Review', Philippe Steer, 03 Nov 2021  reply
I have taken a great pleasure to read the manuscript entitled "A geomorphic-process-based cellular automata model of colluvial wedge morphology and stratigraphy". This manuscript investigates, using the GrainHill cellular automaton model (Tucket et al., 2018, 2020), the role of geomorphological processes in the development and geometrical structure of the sedimentary wedges that develops at the toe of normal fault scarps. The manuscript is overall well written, pleasant to read and, in my opinion, is original by efficiently linking geomorphological laws and processes to the facies and geometry of sediment wedges. It offers some outcomes that could have an impact in geomorphology (dynamics of the scarp-wedge system), but also in paleoseismology (interpretation of wedge structure and facies in terms of earthquake events) and sedimentology (by representing a neat study of the links between geomorphological processes and sedimentological facies). I congratulate the authors for that. In turn, the paper could be published as it is, as I have not noticed any methodological flaws or incorrect interpretation. However, I strongly believe the manuscript could be strengthened by better clarifying and structuring the results using physical dimensionless numbers (see my main comment below), which could also help to reduce the length of the paper (some parts are a bit long). I also give below some minor comments.

I also wish to mention that I am not a sedimentologist and that I was not able to properly assess the quality of the results in terms of sedimentological description.

*Thank you for your time and effort toward our manuscript and for your words of encouragement.*

Main comment:

The results of this manuscript, that are based on numerous models that were performed after a sensitivity analysis in the parameter space (e.g., disturbance rate, lateral collapse rate, weathering rate), could be described in a more structured (and clearer) manner by classifying some main types of models. This would really help to better understand the conditions required to produce different main types of wedges. In this case, the classification is not binary (e.g., transport- or detachment-limited) but considers three main driving processes which are defined by the weathering rate Wo, the disturbance rate D and the lateral collapse rate LCR. Three physical dimensionless numbers could be defined as Wo/D (i.e., the Peclet number already mentioned in the manuscript) to characterize the ability of the model to transport by disturbance (or diffusion) the produced regolith, Wo/LCR to characterize the ability of the model to transport by gravitational processes the produced regolith, and D/LCR to characterize the dominant mode of transport. Each model could then be classified using these three

physical dimensionless numbers and discussed in the light of this classification. I strongly believe that this paper, in particular the Results and Discussion, would benefit from adopting this classified approach, which could really help to clarify the description of the models. In turn, wedges and scarps could be characterized by these numbers, in turn offering a clear physical description of the resulting wedge facies and structure.

This is a great point and we did initially consider casting model results in dimensionless values as you suggest. What may us decide against this approach is our desire to maintain accessibility to non-modelers, particularly more field-based researchers in paleoseismology, Quaternary geology, and sedimentology/pedology. Our hope was to cast the results in the form of 'variable X causes outcome Y' without the extra step of explaining how to interpret a non-dimensionalized value. The goal here is to connect a field-based researchers intuitive understanding of an environmental parameter, say mobile regolith (soil) formation with a theoretical colluvial wedge form and facies distribution. Yes, these results could be explained much more concisely if you assume the audience already has familiarity with model non-dimensionalization. However, I don't think this is necessarily a safe assumption for our target inter-disciplinary audience. However, we did do a basic non-dimensionalization for the ternary diagram shown in the figure below.

I also believe that the main findings could be summer in a synthetic sketch using a kind of ternary-like diagram (with the axes Wo/D, Wo/LCR and D/LCR – even if the sum of these numbers is not necessary one) showing the main types of wedges and scarps.

A ternary diagram to display model results is a really clever idea. Here is what we ended up developing to replace figures 12 (similar figure made for Figure 11):

[Figure]

The axes, as described in the figure caption, are plotted as X/(LCR+W_0+D), where X is the axis variable (LCR,W_0,D). This allows us to plot the model results with the values for each of the three axes summing to 1 for each data point. With your suggested approach, we are able to show both the relationships of colluvial wedge size and relative amounts of wash/debris in a single plot. Thank you for this excellent suggestion.

Minor comments:

Line 38: "we desire this predictive power" – feels a bit strange in a scientific paper

Agreed. Revised to "A robust method to predict colluvial wedge form and facies can develop knowledge toward understanding broader questions…"

Line 39: "do you preserve a post-earthquake colluvial wedge" – could be replaced by "is a post-earthquake colluvial wedge preserved" as the use of "you" seems unusual in a scientific paper.

Agreed. Revised to "under what environmental conditions is a post-earthquake colluvial wedge preserved (or not);"

Line 40: remove the capital letter to "3) How"

Corrected. Thank you.

Line 167: It would be interesting here to mention the typical earthquake magnitude required to generate a 2m tall faut scarp using classical scaling laws between displacement and moment (or magnitude) [Leonard, M. (2010). Earthquake fault scaling: Self-consistent relating of rupture length, width, average displacement, and moment release. Bulletin of the Seismological Society of America, 100(5A), 1971-1988.]

Good idea. However, my understanding is that we would need a value for the rupture length/area to estimate the magnitude correctly. Here we decided to point the reader to an example of a real-world fault system and case study with detailed application of the Leonard (2010) scaling relations. Inserted "In particular, this approximate size scarp often results from typical earthquakes (up to a moment magnitude of ~7) on large normal faults such as the Wasatch Fault in Utah USA (e.g. Bennett et al., 2018)."

Title section 2.2. I suggest rephrasing the title of this section for clarity "Facies definition and transport metrics based on cell tracking" or something else that suits the authors

Agreed. Revised as suggested. Thank you.

Lines 216-220: The choices of the velocity thresholds between the different facies seem rather arbitrary. Could you please justify these values?

Inserted: "The upper threshold for lower debris facies is the approximate average velocity of a hex cell state that travels almost entirely by gravitational free-fall, with a smaller component of movement due to impacts/rebound from other free-falling cells, and thus more likely to be debris." The lower threshold is meant to exclude hex cell states largely traveling by raveling down the wedge slope. Both values are somewhat arbitrary, but allow us to broadly encompass and classified the groupings observed in the scatterplots. Future work should focus on a mechanistic explanation for the observed groupings."

I was also wondering why not using thresholds on the transport time, which intuitively appear as a more natural choice to divide the different facies and has the benefit of displaying some better resolved clusters (on Fig. 8) that would also likely be detected with classical clustering approaches (e.g., dbscan).

The use of transport time is difficult because mobile regolith cells with short transport times includes both those that collapsed off the scarp, with those that were just a part of the pre-earthquake mobile regolith cover with limited periods of disturbance. The

use of velocity captures both the distance and transport time which both affect the resulting sediment facies in real colluvial wedges. The use of dbscan to identify clusters is a good idea. However, I am not sure that an automated clustering method will give insight any further than our simple threshold value method because neither has a direct connection to the mechanics of the sediment transport that is occurring on the scarp.

Lines 244: "both scarps both" – issue with the use of the word "both"

Deleted the second "both".

Figure 8: LCR is not defined in the caption.

Inserted "(LCR)" after "lateral collapse rates".

Figure 11: This figure could go in supplementary as the fact that the volume of the wedge increase with Wo, D and LCR is relatively obvious, given the configuration of the model.

Agreed. Moved to supplemental material as suggested. Thank you.

---

## Author Comment (AC2)

**RC2**: 'Comment on esurf-2021-70', Matan Ben-Asher, 09 Nov 2021  reply

I would first like to thank the authors for this contribution, it was very interesting to read and opened my mind to new ideas. This manuscript attempts to decipher the geomorphic processes that control the evolution of a colluvial wedge following normal faulting and rapid surface deformation. The research of fault scarp evolution has been studied for many years, mostly as a tool to reconstruct paleoseismic activity. The authors use a relatively new physical process based cellular automata numerical model, which is an interesting alternative to the commonly used continuum-style models and by that avoid several limitations and over-simplistic assumptions. Similar models have been recently and successfully used to model various natural landforms, but not fault scarps. The results of this study give new perspective on the evolution of colluvial wedges but also on the much studied and poorly understood processes of soil covered hillslopes evolution.

Thank you for your constructive feedback. We appreciate your time and effort towards our manuscript.

The authors chose not to compare the model with real-world examples but rather run sensitivity analysis using very generalized settings. This limits the application of this study but gives profound basis for further research, which I hope to see in the future.

Yes, you have identified a key strategy of this research. We chose to focus on the broad theoretical grounds of colluvial wedge formation here to fill a knowledge gap on the connections between sedimentary facies and process geomorphology. Our next phase of this work is centered on application and detailed comparison with a specific field site.

The manuscript is well written and original and could be published with minor changed. Below are suggestions that I think would benefit the manuscript, some might require moderate changes and I leave it to the editor and authors to decide if they are required.

Again, thank you for your constructive feedback. It was very helpful and has improved our manuscript.

Main comments

The resulted grouping, as shown in figure 8 is very interesting and deserves a dipper discussion. Several model runs show distinctly different grouping then the ones used (e.g. figure 8 C,G,K). It might hold valuable information on the evolution of colluvial wedge morphology and stratigraphy. It would be very interesting to connect the

observed groups with the different facies, instead of using fixed cutoff values. Reading the manuscript, I was expecting to find a convincing, physically based, argument for the chosen threshold values of the different facies, but did not. I strongly suggest using a simple grouping method, or at least discuss the physical logic in using the same threshold values in mean velocity for different model scenarios.

This is a great point. I (Harrison) could not come up with a physically-based method to classify the groupings for each facies.. My guess is that there potentially is one through statistical mechanics, such as an application of the Fokkar-Plank equation, or through non-local sediment transport theory (e.g. Furbish and Doane, 2021). I beat my head against the wall for a while trying to figure it out, but I think the problem justifies future study beyond the scope of what we present here (i.e. collaboration with someone more capable than I).

As a compromise, I went with estimates using threshold values based on apparent ranges of velocities for a hex cell state in freefall and for one raveling down the wedge slope. I used this simple threshold approach because this did not seem to have any more explanatory power than applying a physics-blind quantitative method, such as a statistical mixing model or cluster grouping method. We now explain this on lines 227-230: " The upper threshold 10 m/day for lower debris facies is the approximate order of magnitude for the velocity of a hex cell state traveling purely by gravity. The lower threshold of 1 m/yr is meant to exclude hex cell states largely traveling by raveling down the wedge. Both values are somewhat arbitrary, but allow us to broadly encompass and classified the groupings observed in the scatterplots. Future work should focus on a mechanistic explanation for the observed groupings."

This simple threshold approach to denote the groupings lets us move on towards answering the questions poised in the introduction, even if theory is not fully complete yet. It will likely take a dedicated study to discover the mechanical underpinnings of the groupings.

The authors chose to use fixed morphological parameters to focus the analysis on geomorphical processes. Looking at the results, it seems that the comparison between 60 and 90 degrees dip shows high sensitivity and results that are sometimes trivial and related to bigger accommodating space and steeper initial dip. It might be better to use fixed dip angle and focus on the process-based parameters, which are less known (covering several orders of magnitudes each) and very challenging to decipher with any other method (unlike dip angle that can be measured directly in a trench).

As we have already done the analysis for the two end-member fault scarp dip angles we prefer to keep the results from both in the paper while including the analysis of the process-based parameters (even if it makes the paper a little long).

According Wallace 1977, which his work is fundamental for this study, the debris-controlled phase ends when the fault scarp reaches the angle of repose. This was also the basis for initial conditions for several models that were used to morphologically date fault scarps. I believe that your results could shed light on the validity of this assumption if addressed more spesifically.

This is actually one of our topics for the future as I believe the topographic profile of the fault scarp requires its own study. To summarize the idea, the debris phase doesn't necessarily end when an angle of repose slope is reached on the wedge, but instead it depends on what the collapse rate is versus the production and disturbance rates. For lower collapse rates, there is not really a debris dominated phase, but rather a mixed phase. The question that needs to be explored is whether natural parent materials are always weak enough to produce debris either during or after an earthquake. We assume that the shaking of an earthquake will always produce the debris but the idea is worth exploring when you can have high indurated materials such as carbonate soils or consolidated bedrock.

Added to the discussion: "Next, the model results provide a physics-based explanation and nuance for the findings of Wallace (1977) who documented fault scarps field exposures across the American West. Wallace (1977) hypothesized that colluvial wedge formation occurs with a debris-dominated phase until the fault scarp free-face is buried, after which a more gradual period of lower energy deposition continues until the eventual burying and topographic smoothening of the whole scarp. Our results show that this hypothesis is supported by physics-based theory when lateral collapse rates are high relative to mobile regolith production and disturbance processes. When lateral collapse rates are comparable or relatively low, Wallace's (1977) hypothesized phases are less distinct, with collapse events occurring stochastically interspersed with periods of disturbance and reworking of wedge material. These interspersed periods of collapse and reworking largely appear to theoretically cause wedge facies stratigraphy to be less distinct than when collapse rates are high. The dependence of wedge facies stratigraphy on process rates provides an explanation for why some colluvial wedge exposures can show classic wedge-shaped forms (e.g. Jackson, 1991; DuRoss et al., 2018), whereas other exposures show less distinct but clearly colluvial wedge deposits (e.g. Bennett et al., 2018). Although not explored here, the topographic evolution of a fault scarp should also have this dependence on process rates, although the effect on fault scarp diffusion dating methods is not yet clear."

I suggest adding another parameter to the analysis - the time that has passed since a particle last moved. It could give valuable information on the timing of facies formation and give a more complete picture of the geomorphic evolution of the colluvial wedge. It would also make results more comparable with luminescence dating data.

Good idea. We have added the suggested parameter to the analysis, including to Figure 7 in the main text and 8 figures showing the value in the parameter space exploration in the supplemental material.

Minor comments

Lines 166-167: How small is the is the initial sediment layer? This should be more methodically defined and described. It is not unlikely that a pre-faulted surface is covered with up to tens of centimeters of mobile regolith. I assume that the thickness of the initial sediment cover could influence results.

Revised to now read: " We create a small 1 cell thick layer of mobile regolith sediment layer to simulate a pre-existing surface soil layer. While a thicker mobile regolith layer may be more realistic for some field sites, a soil/mobile regolith layer's pre-earthquake thickness is tied to the timescales of surface stability and thus a function of earthquake recurrence interval. As noted above, the effects of recurrence interval deserve focused study and thus are not explored here."

Lines 184-185: The classic definition of a peclet number, to my knowledge is different, and the referenced paper is about soil mixing, a process that is not described by the model.

The connection between the reference and the topic is indeed buried. Deep in the discussion section of that paper is an analysis of how the Soil Peclet Number (taken as diffusivity, production, and thickness of soil) appears fairly constant across the planet. With this idea of an approximately constant Soil Peclet Number, we reason that the parameter space of mobile regolith production should share a similar range of orders of magnitude as the mobile regolith diffusivity. This argument is supported by our qualitative observation that the parameter space range we chose appears to cover the range of model outcomes.

Line 185: In figure 11 you show combinations of D and W0 values that range over 3-4 orders of magnitude. To my understanding of your definition of the peclet number (D/W0), this must result on much wider range of values.

Line 189: Why choosing the specific 4 orders of magnitude?

We chose this range because it is the range that is observed in nature as documented by Richardson et al. (2019). Now reads as "We picked these magnitudes from the observed range in mobile regolith diffusivity across the globe (i.e. Richardson et al., 2019)…").

Lines 206-208: It is not clear to me where the threshold value of √3 radians comes from.

Inserted "Note that cells moving purely on an angle-of-repose slope have a transport index of $T\_I=\Delta y/\Delta x=\tan(90°-30°)=\sqrt{3}$ (due to trigonometric right triangle relationships)."

Line 331: Nelson 1992 is repeatedly and rightfully cited in this study, however it is worth addressing the fact that his pioneering work was limited to arid regions.

Good point. Inserted: "Note that Nelson's (1992) facies are based largely on arid region colluvial wedges and variance in facies across climate zones appears likely."

Line 351: I expect that the collapse dominanted stage will end when the surface angle will approach the angle of repose. It would be interesting to test it. It will validate a physical basis of the model and also contribute to studies of long-term modeling of fault scarp evolution that commonly assume rapid evolution until angle of repose is reached.

Added to the discussion on lines 485-498: "Next, the model results provide a physics-based explanation and nuance for the findings of Wallace (1977) who documented fault scarps field exposures across the American West. Wallace (1977) hypothesized that colluvial wedge formation occurs with a debris-dominated phase until the fault scarp free-face is buried, after which a more gradual period of lower energy deposition continues until the eventual burying and topographic smoothening of the whole scarp. Our results show that this hypothesis is supported by physics-based theory when lateral collapse rates are high relative to mobile regolith production and disturbance processes. When lateral collapse rates are comparable or relatively low, Wallace's (1977) hypothesized phases are less distinct, with collapse events occurring stochastically interspersed with periods of disturbance and reworking of wedge material. These interspersed periods of collapse and reworking largely appear to theoretically cause wedge facies stratigraphy to be less distinct than when collapse rates are high. The dependence of wedge facies stratigraphy on process rates provides an explanation for why some colluvial wedge exposures can show classic wedge-shaped forms (e.g. Jackson, 1991; DuRoss et al., 2018), whereas other exposures show less distinct but clearly colluvial wedge deposits (e.g. Bennett et al., 2018). Although

not explored here, the topographic evolution of a fault scarp should also have this dependence on process rates, although the effect on fault scarp diffusion dating methods is not yet clear."

Technical corrections

Lines 103-104: Concider citing BenDror and Goren, 2018, JGR: Earth Surface.

That is a good paper to cite here. Inserted. Thank you.

Line 153: I believe it was Culling, 1963 who first used the term 'diffusivity' in soil transport.

Added a citation to Culling, 1963

Line 215: add reference to figure 8.

Added a reference to Figure 8.

Line 340: Give references to these observations.

Revised to "The similarity between the facies in the model and sedimentary facies provides a mechanistic explanation for the complex stratigraphy observed in actual colluvial wedges (Figures 2, 9, 10)."

Again, thank you for your time and effort towards our manuscript.

---

## Author Response (AR2)

28 Jan 2022
**Editor decision: Publish subject to technical corrections**
by Niels Hovius
**Comments to the author**:
Dear Harrison Gray and colleagues,
I am happy to follow the recommendation of associate editor Greg Hancock, and approve publication of your manuscript in ESurf pending technical corrections to grammar and spelling. I ask that you give this issue your careful attention before submitting your finalized materials. The copy editors of Copernicus can assist with this task, but I trust that have the means to address most if not all niggles.
The manuscript will now be handled by the Copernicus publishers. Your timely and efficient response to their prompts will help to keep time to print minimal.
Thank you for submitting youuur excellent work to ESurf, and for working with our AE to further improve the manuscript. I look forward to seeing your work published in its final form and trust that it will receive good attention.
Niels Hovius

Thank you for your review of our manuscript. We have gone through and made small grammar and spelling fixes as needed.

27 Jan 2022
**Associate Editor decision: Publish as is**
by Greg Hancock
**Comments to the author**:
The Authors have submitted an excellent paper for review which is recognised by the Reviewers. The paper is ready for publication. However, please do a final check for English, grammar and spelling.
I look forward to seeing the paper in print

Thank you for handling the review of our manuscript. We have gone through and made small grammar and spelling fixes as needed.